# Building Change Detection with Deep Learning by Fusing Spectral and Texture Features of Multisource Remote Sensing Images: A GF-1 and Sentinel 2B Data Case

Junfu Fan [1,2], Mengzhen Zhang [1,2], Jiahao Chen [1,3,*], Jiwei Zuo [1], Zongwen Shi [1] and Min Ji [4]

[1] School of Civil Engineering and Geomatics, Shandong University of Technology, Zibo 255000, China; fanjf@sdut.edu.cn (J.F.); zhangmz@lreis.ac.cn (M.Z.); 21407010773@stumail.sdut.edu.cn (J.Z.); 21407010766@stumail.sdut.edu.cn (Z.S.)

[2] State Key Laboratory of Resources and Environmental Information System, Institute of Geographical Sciences and Natural Resources Research, Chinese Academy of Sciences, Beijing 100101, China

[3] College of Urban and Environmental Sciences, Central China Normal University, Wuhan 430079, China

[4] College of Geodesy and Geomatics, Shandong University of Science and Technology, Qingdao 266510, China; jimin@sdust.edu.cn

* Correspondence: chenjh@mails.ccnu.edu.cn

**Abstract:** Building change detection is an important task in the remote sensing field, and the powerful feature extraction ability of the deep neural network model shows strong advantages in this task. However, the datasets used for this study are mostly three-band high-resolution remote sensing images from a single data source, and few spectral features limit the development of building change detection from multisource remote sensing images. To investigate the influence of spectral and texture features on the effect of building change detection based on deep learning, a multisource building change detection dataset (MS-HS BCD dataset) is produced in this paper using GF-1 high-resolution remote sensing images and Sentinel-2B multispectral remote sensing images. According to the different resolutions of each Sentinel-2B band, eight different multisource spectral data combinations are designed, and six advanced network models are selected for the experiments. After adding multisource spectral and texture feature data, the results show that the detection effects of the six networks improve to different degrees. Taking the MSF-Net network as an example, the F1-score and IOU improved by 0.67% and 1.09%, respectively, compared with high-resolution images, and by 7.57% and 6.21% compared with multispectral images.

**Keywords:** building change detection; deep learning; high-resolution; multispectral; multisource spectral data

## 1. Introduction

Building change detection is an important research topic in the field of remote sensing, which refers to the design of relevant algorithms to extract the building change characteristics in different periods of images in the same area [1], which plays an important role in the investigation of damaged buildings and the study of urbanization development processes [2,3]. The development of remote sensing technology has led to the continuous improvement of image quality [4]; high-resolution images have fine texture features and multispectral images are rich in spectral features. Combining these two types of images and investigating their effects on the building change detection is particularly important for further expansion of the field.

Traditional methods of change detection [5] are principal component analysis [6], arithmetic operations [7,8], etc. These methods are simple to operate and only require the arithmetic calculation of remote sensing images in different periods, but their detection effect is poor and they cannot realize batch processing of images, which is difficult to deal with the increasing surge in massive remote sensing data [4,9,10]. The rapid development

of deep learning [11], especially the powerful image processing capability of convolutional neural networks (CNNs) [12], has led to the wide use of CNNs in the remote sensing field [13,14], including building change detection from remote sensing imagery [15,16].

Early fusion networks such as fully convolutional early fusion (FC-EF) [17] based on fully convolutional networks (FCNs) [18] and UNet++_MSOF [19] based on UNet++ [20] have been proposed successively, and scholars have achieved results by inputting dual-temporal remote sensing images in series to the network for change detection. Daudt et al. proposed the FC-Siam-conc and FC-Siam-diff networks [17], which combine FCN and Siamese networks [21] for change detection. This method can learn the deep-level features of a single remote sensing image and improve detection accuracy. Since then, Siamese networks such as NestNet [10], Siamese Nested UNet [22], DASNet [23], and PGA-SiamNet [24] have been proposed to further develop the remote sensing change detection field. STA-Net [25], ADS-Net [26], and MSF-Net [9] for building change detection in remote sensing images have also been proposed to improve building change detection accuracy.

In addition to the Siamese network structures, scholars have added attention mechanisms [27,28] to change detection networks. Zhang et al. designed the deeply supervised image fusion network (DSIFN) [4], which introduces a spatial attention mechanism [29] and channel attention mechanism [30] to effectively improve change feature detection. Chen et al. designed STA-Net [25] to introduce a spatial-temporal attention mechanism to the network to capture long-term spatial-temporal dependencies to learn better building features. Wang et al. designed ADS-Net [26], adding a convolutional block attention module [31] to the network to reconstruct features for multiscale information and improve the building change detection ability. Chen et al. designed MSF-Net [9], introducing selective kernel convolution [32] and channel and spatial multiple attention mechanisms to fuse multiscale features for building change extraction.

Although building change detection networks have matured in development, current research is mostly based on single-source high-resolution image datasets (e.g., LEVIR-CD [25], WHU Building Dataset [33], AIST Building Change Detection [34], SYSU-CD dataset [35]) and single-source multispectral image datasets (OSCD dataset [36]). Single satellite sensors have specific revisit periods for the same area, are affected by weather, such as clouds and fog, and cannot provide continuous high-quality image data in the same area [37]. This condition limits ground-surface dynamic monitoring tasks such as change detection, so it is necessary to conduct research on building change detection with multiple data sources. For remote sensing imagery, the high-resolution satellite-based datasets, as described above, have higher spatial resolution and rich texture features, but their spectral features are insufficient. Multispectral satellite-based datasets have rich spectral features, but their spatial resolution is lower, the texture features are insufficient, and the existing OSCD multispectral dataset containing building change information only has 24 image pairs of data, which poses a limitation to the training effect of the neural network. Effectively combining multisource remote sensing images and using more textural and spectral features plays an important role in improving building change detection accuracy.

For multisource change detection, Liu proposed a change discovery and update method for high-consequence areas based on multisource remote sensing imagery data [38] using Landsat-8 multispectral images to assist with high-resolution imaging for change detection and achieve higher accuracy results. Zhao proposed a land use change detection method based on multisource data, combining imagery and vector data to achieve fully automatic and efficient land use change detection and extraction [39]. Wang conducted change detection experiments based on a convolutional neural network and multisource high-resolution remote sensing data of ZY-3 and GF-2 to design a hybrid convolutional feature extraction module, a hybrid interleaved group convolution module, and a multi-loss supervised training method to obtain fine change detection results [37]. Zhang et al. designed W-Net [40], which can be used for change detection tasks of single-source and multisource remote sensing data. The experiments showed that combined multisource data

as the model input, which combines the advantages of spectral, texture, and, structural information, can significantly improve the robustness of the model. Chen et al. designed the deep Siamese convolutional multiple-layer recurrent neural network (SiamCRNN) [41], which can perform change detection tasks with heterogeneous source images in front and behind time phases. Seydi et al. proposed an end-to-end multidimensional CNN framework [42] for land use change detection in multisource remote sensing data using three different types of remote sensing datasets (multispectral, hyperspectral, and polarized synthetic aperture radar) to evaluate the effectiveness and reliability of the proposed method.

From the above research, it can be concluded that: (1) the existing building change detection datasets are mostly single-source high-resolution images and single-source multispectral images, and there is a lack of open-source multisource remote sensing image-based datasets. High-resolution datasets such as LEVIR-CD are rich in texture features but insufficient in spectral information, while multispectral datasets such as OSCD are sufficient in spectral information but insufficient in texture features, and the too small data size of this dataset limits the learning capability of the model. Therefore, the lack of multisource building change detection dataset limits the further development of this field. (2) The current research on multisource data change detection focuses on designing a change detection method that can be used for multiple data sources. However, there are few comparative studies on the impact of adding multisource spectral features and texture features on improving detection accuracy; the main reason for this problem is also due to the lack of datasets.

To solve the above problems, this paper uses GF-1 high-resolution images and Sentinel-2B multispectral imageries, proposes a multisource remote sensing image building change detection dataset (MS-HS BCD dataset), and provides a database for studying multisource building change detection; our proposed dataset will be released through GitHub (https://github.com/arcgislearner/MS-HS-BCD-dataset (accessed on 6 April 2023)). Additionally, based on the MS-HS BCD dataset, six open-source state-of-the-art change detection network models are selected to explore the effects of fusing multisource spectral features and texture features on building change detection effects. The experimental results show that after fusing the multisource texture and spectral information, the detection effect of all six network models improved compared with that of a single data source. The proposed dataset and study provide a database for multisource building change detection research and a reference for further research applications in this field.

## 2. Study Area and Data Processing

### 2.1. Study Area

In this paper, dual-temporal images of Huangdao District, Qingdao City, Shandong Province, were selected to produce a multisource image building change detection dataset. Huangdao District is located in the southeastern part of the Shandong Peninsula, near the Yellow Sea, with latitude 35°35′~36°08′N and longitude 119°30′~120°11′E (as shown in Figure 1) and is the ninth national new district of the People's Republic of China. In recent years, the Huangdao District experienced rapid economic development, accelerated urbanization, and continuous reconstruction of the old city. This has resulted in significant change in the types of buildings, which provides a possibility for producing the building change detection dataset in this paper.

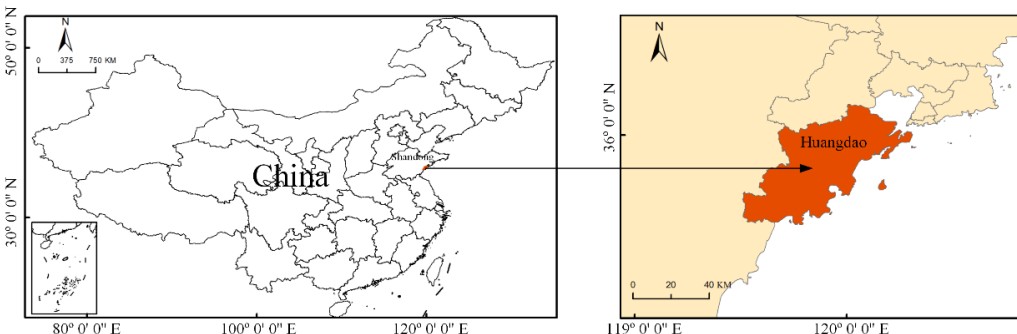

**Figure 1.** Study area.

### 2.2. Data Selection and Preprocessing

In this study, GF-1 high-resolution remote sensing images and Sentinel-2B multispectral remote sensing images from February 2019 and December 2019 were used to produce the dataset. The GF-1 data selected for this paper were obtained from Shandong University of Science and Technology; due to the limitation of the data acquisition party, it can only be known that the GF-1 data were obtained in February and December. The Sentinel-2B data were obtained from ESA Copernicus Data Center (https://scihub.copernicus.eu/dhus/#/home (accessed on 12 April 2021)), and the specific acquisition dates for the two Sentinel-2B are 16 February 2019 and 7 December 2019. To ensure the reliability of the production of the dataset, the buildings in the GF-1 and Sentinel-2B images in February and December were manually checked to be consistent. The GF-1 images are RGB 3-band with 2 m/pixel spatial resolution after image fusion, and Sentinel-2B data are 13-band images with spatial resolutions of 10, 20, and 60 m/pixel. The parameters of the two satellite images are shown in Tables 1 and 2.

**Table 1.** Detailed parameters of GF-1 bands.

| Band | Band Name | Spatial Resolution (m/Pixel) | Wavelength (μm) |
|---|---|---|---|
| 1 | Blue | 2 | 0.52–0.59 |
| 2 | Green | 2 | 0.63–0.69 |
| 3 | Red | 2 | 0.77–0.89 |

**Table 2.** Detailed parameters of Sentinel-2B bands.

| Band | Band Name | Spatial Resolution (m/Pixel) | Wavelength (μm) |
|---|---|---|---|
| 1 | Coastal Aerosol | 60 | 0.43–0.45 |
| 2 | Blue | 10 | 0.46–0.52 |
| 3 | Green | 10 | 0.54–0.58 |
| 4 | Red | 10 | 0.65–0.68 |
| 5 | Vegetation Red Edge | 20 | 0.70–0.71 |
| 6 | Vegetation Red Edge | 20 | 0.73–0.75 |
| 7 | Vegetation Red Edge | 20 | 0.77–0.79 |
| 8 | NIR | 10 | 0.79–0.90 |
| 8A | Vegetation Red Edge | 20 | 0.85–0.88 |
| 9 | Water Vapor | 60 | 0.94-0.96 |
| 10 | SWIR-Cirrus | 60 | 1.36–1.39 |
| 11 | SWIR | 20 | 1.57–1.66 |
| 12 | SWIR | 20 | 2.10–2.28 |

In this paper, the GF-1 images were preprocessed with radiometric correction and georeferencing, and only image resampling was performed. The maximum spatial resolution of the Sentinel-2B multispectral image is 10 m/pixel. To facilitate the cropping of the corresponding areas of the two images and inspection of the features in the corresponding photos after cropping, the spatial resolution of the GF-1 image was resampled

to 2.5 m/pixel using the resampling tool in ArcGIS, and the resampling method was the nearest neighbor method. The Sentinel-2B images were obtained at the L1C level with orthorectification and geometric correction. To produce the L2A level data, the L1C level data were radiometrically calibrated and atmospherically corrected using the Sen2cor plug-in released by ESA. The L2A level data were processed to remove the 10th band of data, leaving 12 bands. The image of the L2A level was then resampled using the Sen2cor tool to upgrade the 20 m/pixel and 60 m/pixel spatial resolution bands to 10 m/pixel spatial resolution, and each data band was saved as a TIFF file. Finally, all bands of the L2A Sentinel-2B images were resampled to 2.5 m/pixel spatial resolution also using the resampling tool in ArcGIS; the resampling method was the nearest neighbor method, so that they had the same resolution as that of GF-1 images, which was convenient for subsequent data processing and neural network training. We also used the geographic registration tool in ArcGIS software to carry out geographic registration of GF-1 and Sentinel-2B in two periods, respectively. The registration method used was affine transformation. Due to the limitation of resolution, the registration error was maintained at about half a pixel.

*2.3. Dataset Production*

ArcGIS is used in this paper to label buildings that changed in two periods, labeling objects such as buildings, movable houses, and agricultural sheds, with a total of 3646 elements. The annotation files were exported to raster data files, and the pixel points of changed buildings were marked as one; those of unchanged buildings were marked as zero using a raster calculator. Considering the computing power of the GPU, the annotation file, the 3-band GF-1 images, and the 12 single-band image maps of the processed L2A-level Sentinel-2B were cropped into 256 × 256, and the cropped annotation images were checked for building change annotation to modify mislabeled and omitted objects. To reduce the negative impact of the imbalance between the changed and unchanged samples on the model training, the cropped image pairs without building changes were excluded, and a total of 600 sets of building change images were obtained. Each set of images contains the prechange image, the postchange image, and the change annotation file. The pre- and postchange images were composed of the GF-1 images and the 12 single-band Sentinel-2B images.

After a series of operations such as resampling and cropping of Sentinel-2B multispectral images, the pixel depth of single-band data was 16-bit unsigned data type, taking values from 0 to 65,535, and some pixel values in the image area reached more than 7000. If the image data are directly loaded into the neural network training, the excessive value will make the loss drop unstable and increase the difficulty of neural network model training. Therefore, this study converted the image data with a pixel depth of 16-bit unsigned data type into 8-bit unsigned data type with a range of values from 0 to 255.

In this paper, the finally generated 600 image sets were divided into 540 sets of training images, 30 sets of validation images, and 30 sets of test images. This multisource image building change detection dataset was named the multispectral-high-resolution building change detection dataset (MS-HS BCD dataset). Dataset production and the experimental flow of this paper are shown in Figure 2.

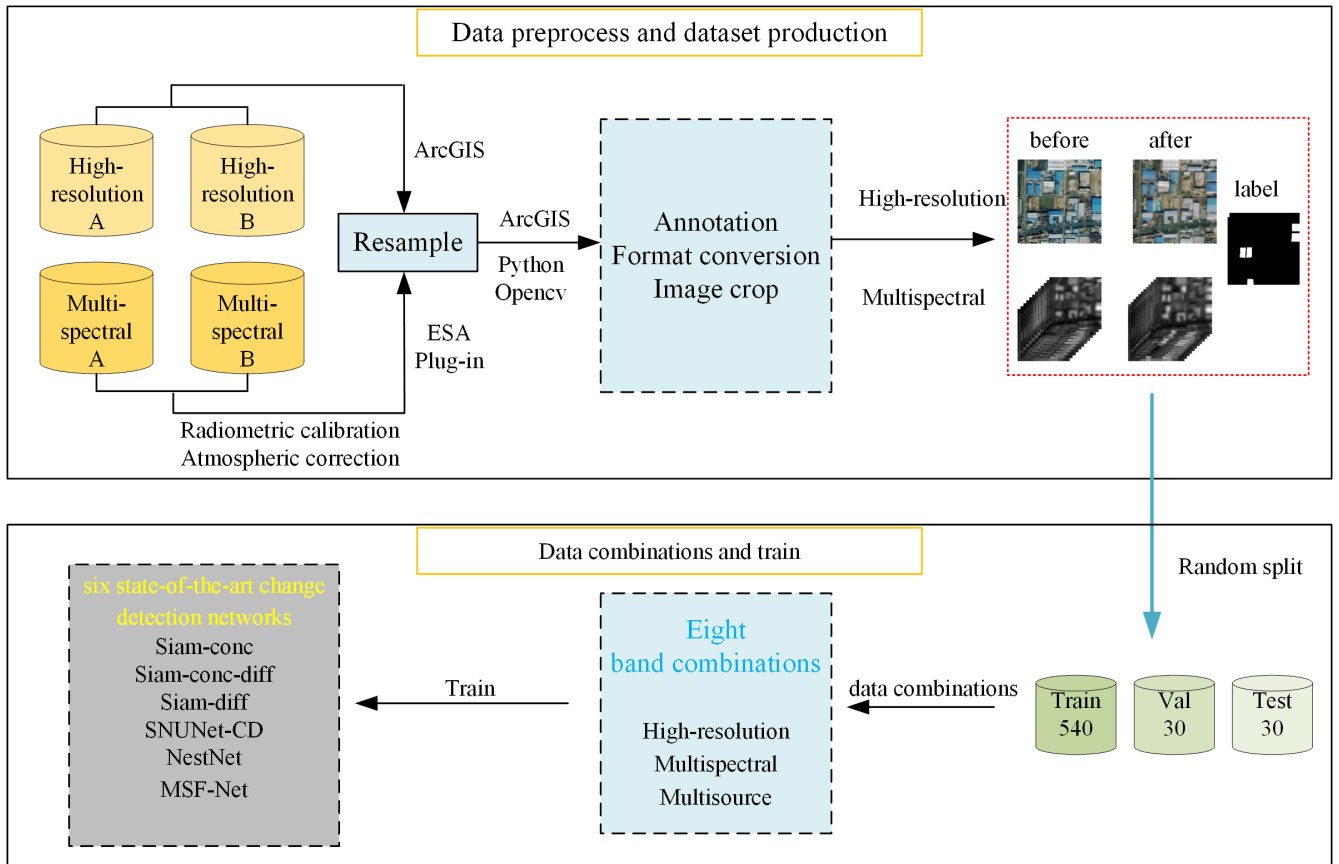

**Figure 2.** Flowchart of the method used in this study.

### 3. Methods

#### 3.1. Deep Neural Network for Building Change Detection

In this paper, six open-source, state-of-the-art or widely used change detection networks, MSF-Net [9], Siam-conc, Siam-diff, Siam-conc-diff [22], SNUNet-CD [43], and NestNet [10], are selected to explore the impact of multisource spectral data on building change detection improvement.

1. MSF-Net: A state-of-the-art multiscale supervised fusion network based on attention mechanisms; the network structure is shown in Figure 3. MSF-Net built dual-context fusion module (Figure 3b) to obtain global context information of buildings, introduced channel attention mechanism (Figure 3e), selective kernel convolution (Figure 3f) to the network encoding (Figure 3a), and decoding (Figure 3d) modules to enhance the building change detection capability. A new multiscale fusion module and multiscale output module are designed to enable the network model to simultaneously extract buildings at different scales. The powerful feature extraction capability and the state-of-the-art nature are the reasons why we use it in this paper.

The network structure of Siam-conc, Siam- diff, and Siam-conc-diff are shown in Figure 4. These three network structures are a combination of a Siamese structure and Unet++. This combination is widely used in the field of change detection, where the Unet++ network model improves the multiscale detection capability and the Siamese structure enables the model to simultaneously learn deeper building features in the dual-temporal images, effectively improving the building change detection accuracy.

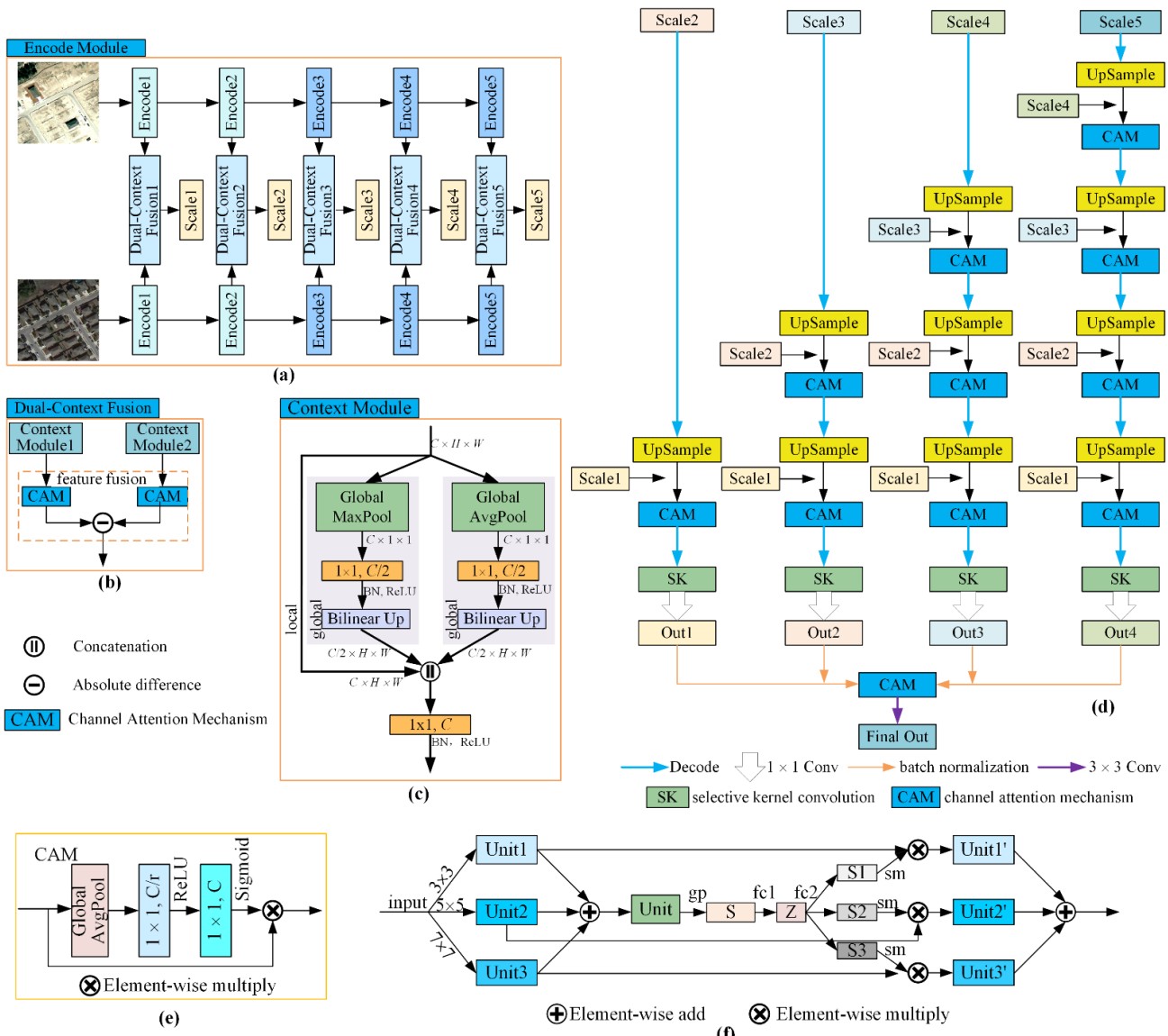

**Figure 3.** The network structure of MSF-Net. (**a**) basic structure of encoding module, (**b**) dual-context fusion module, (**c**) context module, (**d**) basic structure of decoding module, (**e**) channel attention mechanism, (**f**) selective kernel convolution.

2. Siam-conc: As shown in Figure 4, in the Siam-conc network, the "Operation" is channel concatenate; Siamese UNet++ connects the channels of the before and after change images in series for the building change detection.

3. Siam-diff: In the Siam-diff network, the "Operation" in Figure 4 is used to calculate the difference between two images before and after the change, then Siamese UNet++ feeds the calculated difference into the next network structure to detect the changed building.

4. Siam-conc-diff: In the Siam-conc-diff network, the "Operation" in Figure 4 is used to perform a channel concatenation operation on the images before and after the change and the result of its differential operation, then Siamese UNet++ performs feature extraction on the concatenation result to detect the changed buildings.

5. SNUNet-CD: The SNUNet-CD network structure is shown in Figure 5. An improved Siamese UNet++ uses the Ensemble Channel Attention Module (ECAM, Figure 5b) to combine the outputs of multiple branches into one output to obtain representative features at different scales. This process can improve the building change detection

accuracy at different scales. The advanced nature of this network structure makes it widely cited, which is why we chose it for our study.

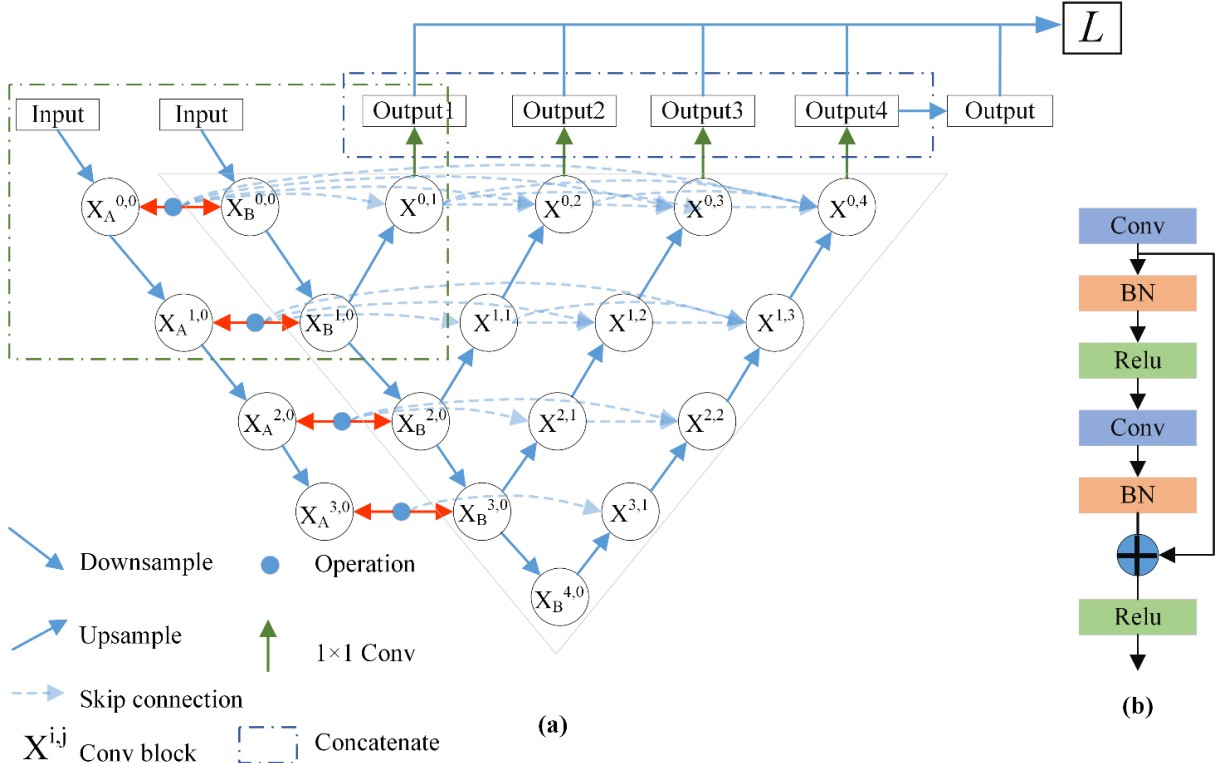

**Figure 4.** The network structure of Siam-conc, Siam- diff, Siam-conc-diff. (**a**) basic structure of network, (**b**) detailed structure of the Conv block.

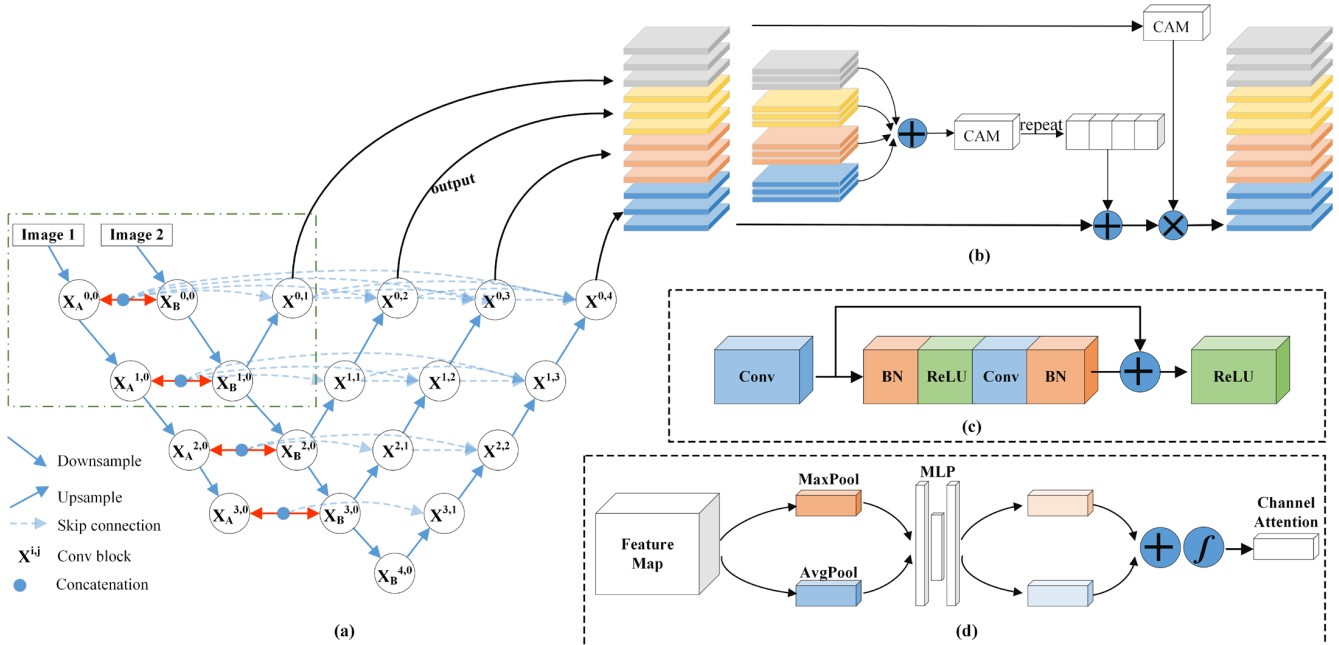

**Figure 5.** The network structure of SNUNet-CD. (**a**) backbone of SNUNet-CD, (**b**) Ensemble Channel Attention Module, (**c**) Convolution Block, (**d**) Channel Attention Module (CAM).

6. NestNet: The network structure of NestNet is shown in Figure 6. NestNet improves the dense skip connection module based on Siamese Unet++ and uses the difference absolute value operation to process remote sensing images and learn building change features at multiple scales. The effective network structure makes it have an excellent detection effect, and the open-source and advanced features are also the reasons why we select it as our research model.

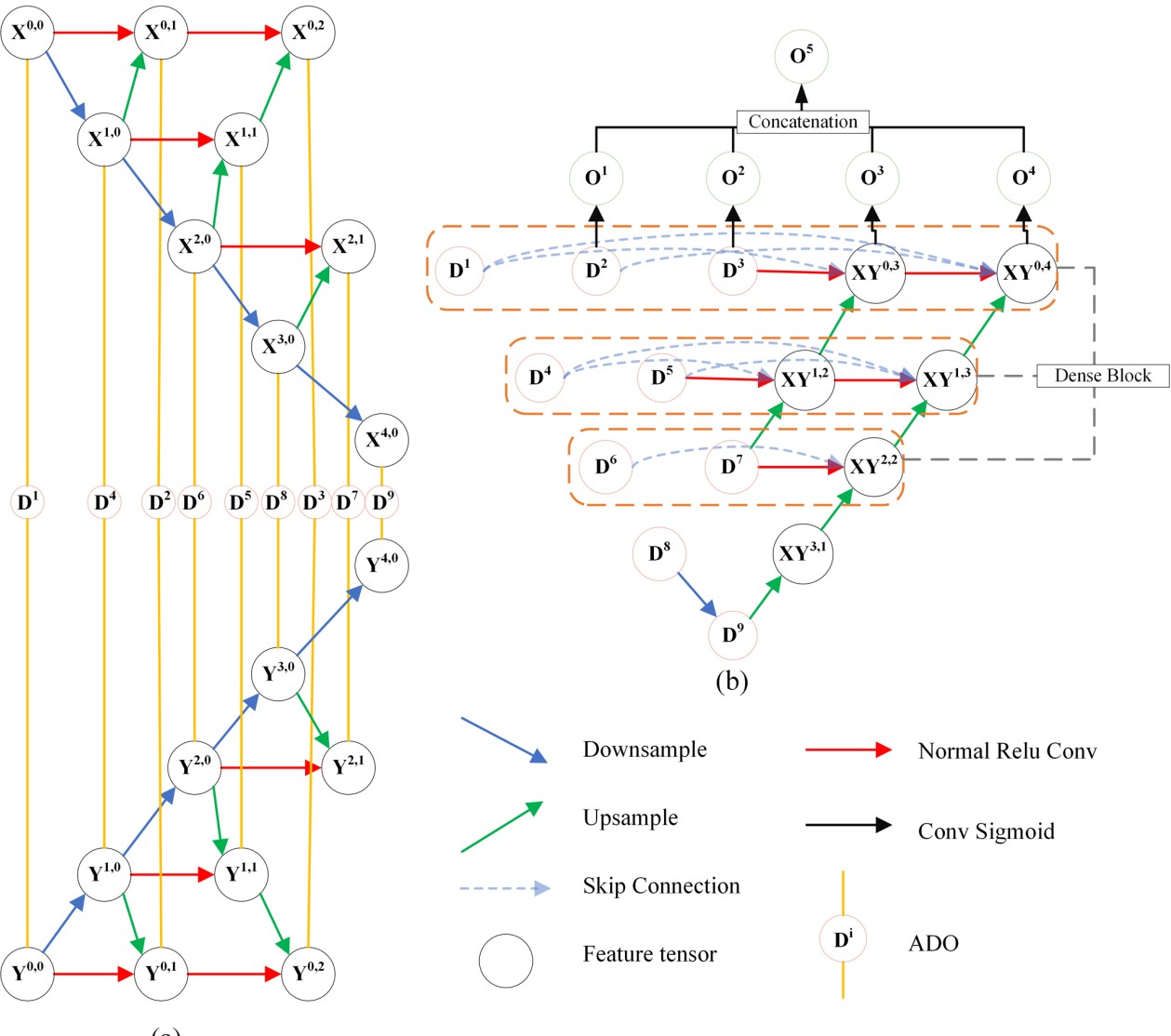

**Figure 6.** The network structure of NestNet. (**a**) Downsampling module, (**b**) Upsampling module.

### 3.2. Combination of MS-HS BCD Datasets

To investigate the building change detection of multisource spectral and texture feature data, based on the MS-HS BCD dataset, we set up eight data combinations for the training set, validation set, and test set used for the deep neural network in this paper. One combination is only the single-source GF-1 high-resolution imageries with three bands (red, green, blue), as shown in Table 3. Four combinations are only the single-source Sentinel-2B multispectral images, according to the difference of original resolution before the resampling of Sentinel-2B, as shown in Tables 4–7. Other combinations include the multisource image pair that combines high-resolution images with multispectral images. Again, depending on the original Sentinel-2B resolution, there are three combinations which are shown in Tables 8–10. Table 8 shows the combination of RGB bands in the GF-1

image and band NIR with the original resolution of 10 m/pixel in the Sentinel-2B image. Table 9 shows the basis of the band combination in Table 8 with added bands of Vegetation Red Edge and SWIR with the original resolution of 20 m/pixel. Table 10 shows, on the basis of the band combination in Table 9, that the bands of Coastal Aerosol and Water Vapor were added with an original resolution of 60 m/pixel.

**Table 3.** High-resolution data combination.

| Band Number | Band Source | Wavelength (μm) |
|:---:|:---:|:---:|
| 1 | GF-1 Blue | 0.52–0.59 |
| 2 | GF-1 Green | 0.63–0.69 |
| 3 | GF-1 Red | 0.77–0.89 |

**Table 4.** The first combination of the single-source Sentinel-2B multispectral images.

| Band Number | Band Source | Wavelength (μm) |
|:---:|:---:|:---:|
| 1 | Sentinel-2B Blue | 0.46–0.52 |
| 2 | Sentinel-2B Green | 0.54–0.58 |
| 3 | Sentinel-2B Red | 0.65–0.68 |

**Table 5.** The second combination of the single-source Sentinel-2B multispectral images.

| Band Number | Band Source | Wavelength (μm) |
|:---:|:---:|:---:|
| 1 | Sentinel-2B Blue | 0.46–0.52 |
| 2 | Sentinel-2B Green | 0.54–0.58 |
| 3 | Sentinel-2B Red | 0.65–0.68 |
| 4 | Sentinel-2B NIR | 0.79–0.90 |

**Table 6.** The third combination of the single-source Sentinel-2B multispectral images.

| Band Number | Band Source | Wavelength (μm) |
|:---:|:---:|:---:|
| 1 | Sentinel-2B Blue | 0.46–0.52 |
| 2 | Sentinel-2B Green | 0.54–0.58 |
| 3 | Sentinel-2B Red | 0.65–0.68 |
| 4 | Sentinel-2B Vegetation Red Edge | 0.70–0.71 |
| 5 | Sentinel-2B Vegetation Red Edge | 0.73–0.75 |
| 6 | Sentinel-2B Vegetation Red Edge | 0.77–0.79 |
| 7 | Sentinel-2B NIR | 0.79–0.90 |
| 8 | Sentinel-2B Vegetation Red Edge | 0.85–0.88 |
| 9 | Sentinel-2B SWIR | 1.57–1.66 |
| 10 | Sentinel-2B SWIR | 2.10–2.28 |

**Table 7.** The fourth combination of the single-source Sentinel-2B multispectral images.

| Band Number | Band Source | Wavelength (μm) |
|:---:|:---:|:---:|
| 1 | Sentinel-2B Coastal Aerosol | 0.43–0.45 |
| 2 | Sentinel-2B Blue | 0.46–0.52 |
| 3 | Sentinel-2B Green | 0.54–0.58 |
| 4 | Sentinel-2B Red | 0.65–0.68 |
| 5 | Sentinel-2B Vegetation Red Edge | 0.70–0.71 |
| 6 | Sentinel-2B Vegetation Red Edge | 0.73–0.75 |
| 7 | Sentinel-2B Vegetation Red Edge | 0.77–0.79 |
| 8 | Sentinel-2B NIR | 0.79–0.90 |
| 9 | Sentinel-2B Vegetation Red Edge | 0.85–0.88 |
| 10 | Sentinel-2B Water Vapor | 0.94–0.96 |
| 11 | Sentinel-2B SWIR | 1.57–1.66 |
| 12 | Sentinel-2B SWIR | 2.10–2.28 |

**Table 8.** Multisource data four-band combination.

| Band Number | Band Source | Wavelength (μm) |
| --- | --- | --- |
| 1 | GF-1 Blue | 0.52–0.59 |
| 2 | GF-1 Green | 0.63–0.69 |
| 3 | GF-1 Red | 0.77–0.89 |
| 4 | Sentinel-2B NIR | 0.79–0.90 |

**Table 9.** Multisource data ten-band combination.

| Band Number | Band Source | Wavelength (μm) |
| --- | --- | --- |
| 1 | GF-1 Blue | 0.52–0.59 |
| 2 | GF-1 Green | 0.63–0.69 |
| 3 | GF-1 Red | 0.77–0.89 |
| 4 | Sentinel-2B Vegetation Red Edge | 0.70–0.71 |
| 5 | Sentinel-2B Vegetation Red Edge | 0.73–0.75 |
| 6 | Sentinel-2B Vegetation Red Edge | 0.77–0.79 |
| 7 | Sentinel-2B NIR | 0.79–0.90 |
| 8 | Sentinel-2B Vegetation Red Edge | 0.85–0.88 |
| 9 | Sentinel-2B SWIR | 1.57–1.66 |
| 10 | Sentinel-2B SWIR | 2.10–2.28 |

**Table 10.** Multisource data twelve-band combination.

| Band Number | Band Source | Wavelength (μm) |
| --- | --- | --- |
| 1 | GF-1 Blue | 0.52–0.59 |
| 2 | GF-1 Green | 0.63–0.69 |
| 3 | GF-1 Red | 0.77–0.89 |
| 4 | Sentinel-2B Coastal Aerosol | 0.43–0.45 |
| 5 | Sentinel-2B Vegetation Red Edge | 0.70–0.71 |
| 6 | Sentinel-2B Vegetation Red Edge | 0.73–0.75 |
| 7 | Sentinel-2B Vegetation Red Edge | 0.77–0.79 |
| 8 | Sentinel-2B NIR | 0.79–0.90 |
| 9 | Sentinel-2B Vegetation Red Edge | 0.85–0.88 |
| 10 | Sentinel-2B Water Vapor | 0.94–0.96 |
| 11 | Sentinel-2B SWIR | 1.57–1.66 |
| 12 | Sentinel-2B SWIR | 2.10–2.28 |

*3.3. Experiment Environment*

The eight dataset combinations designed in this paper are inputted to the six selected building change detection models for training and validation. During the training process of the network, data augmentation of the training dataset is performed using methods such as random rotate, random noise, and random flip to reduce the adverse effects of small datasets on network training and to enhance the robustness of the network. The experiment environment is shown in Table 11; all of the network models are built based on the PyTorch deep learning framework, the programming language is Python, the programming environment is PyCharm, and the training parameters of each network model are consistent with the original paper. The experiments are run on a workstation with an AMD Ryzen 9 5950X 16-core (3.4 GHz) CPU, 128 GB RAM, and an Nvidia GeForce RTX 3090 (24 GB) GPU.

**Table 11.** The specific experiment environment.

| Type | Environment | Detail |
|------|-------------|--------|
| software | Framework | PyTorch |
|  | Language | Python |
|  | Programming | PyCharm |
| hardware | CPU | AMD Ryzen 9 5950X 16-core (3.4 GHz) |
|  | GPU | Nvidia GeForce RTX 3090 (24 GB) |
|  | RAM | 128 GB |

*3.4. Evaluation Metric*

In this paper, four metrics, precision (*P*), recall (*R*), F1-score (*F1*), and intersection over union (*IOU*), are selected to verify the effectiveness of the proposed model. The higher the precision is, the more the model detects the correct change pixels. The higher the recall is, the more the model detects the correct change pixels. The higher the F1-score and *IOU* are, the better the overall performance of the model. The calculations of the four metrics are shown below:

$$P = \frac{TP}{TP + FP} \tag{1}$$

$$R = \frac{TP}{TP + FN} \tag{2}$$

$$F1 = \frac{2P \times R}{P + R} \tag{3}$$

$$IOU = \frac{TP}{TP + FP + FN} \tag{4}$$

where *TP* represents pixels that actually changed and were predicted to change by the model, *TN* represents pixels that actually did not change and were predicted to not change by the model, *FP* represents pixels that actually did not change but were predicted to change by the model, and *FN* represents pixels that actually changed but were predicted to not change by the model.

**4. Results**

*4.1. Single-Source High-Resolution Remote Sensing Images Building Change Detection*

Based on multiple dataset combinations in the MS-HS BCD dataset, this paper first performs change detection of single-source high-resolution remote sensing images, with the data source being the three-band GF-1 RGB images (Table 3). The training loss curves for the six network structures are shown in Figure 7; the losses of the six network models remained flat after epoch50, with NestNet having the highest loss and SNUNet-CD having the lowest loss. The detection results of the six network models are shown in Table 12 and Figure 8.

From Table 12, the experimental results show that the MSF-Net algorithm performs the best, with the four metrics of precision, recall, F1-score, and IOU reaching 61.1%, 65.02%, 58.55%, and 43.31%, respectively. These results are 11.03%, 10.25%, 10.53%, and 8.24%, respectively, higher than those of the Siam-diff algorithm, which performs the best among the other advanced model algorithms. The worst performing algorithm is NestNet, with an F1-score and IOU of only 43.84% and 30.96%, respectively, but its recall index improves significantly compared with other network models, reaching 60.46%.

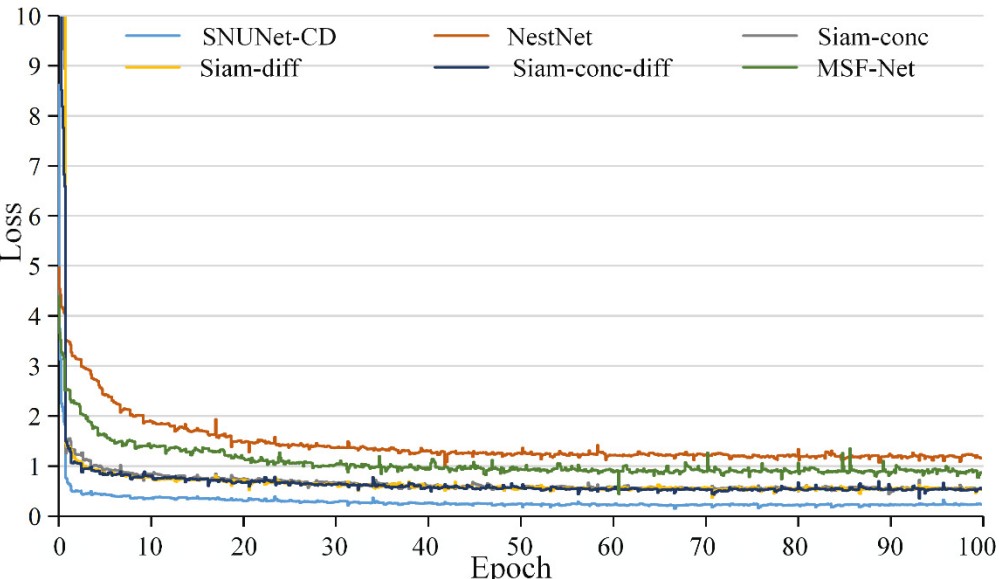

**Figure 7.** Training loss of single-source high-resolution images.

**Table 12.** Building change detection results of single-source high-resolution images.

| Model | Precision | Recall | F1-Score | IOU |
|---|---|---|---|---|
| Siam-conc | 48.43% | 53.29% | 45.3% | 31.49% |
| Siam-conc-diff | 54.12% | 53.25% | 47.06% | 33.99% |
| Siam-diff | 50.08% | 54.77% | 48.02% | 35.07% |
| SNUNet-CD | 43.42% | 55.22% | 44.68% | 31.83% |
| NestNet | 41.48% | 60.46% | 43.84% | 30.96% |
| MSF-Net | 61.1% | 65.02% | 58.55% | 43.31% |

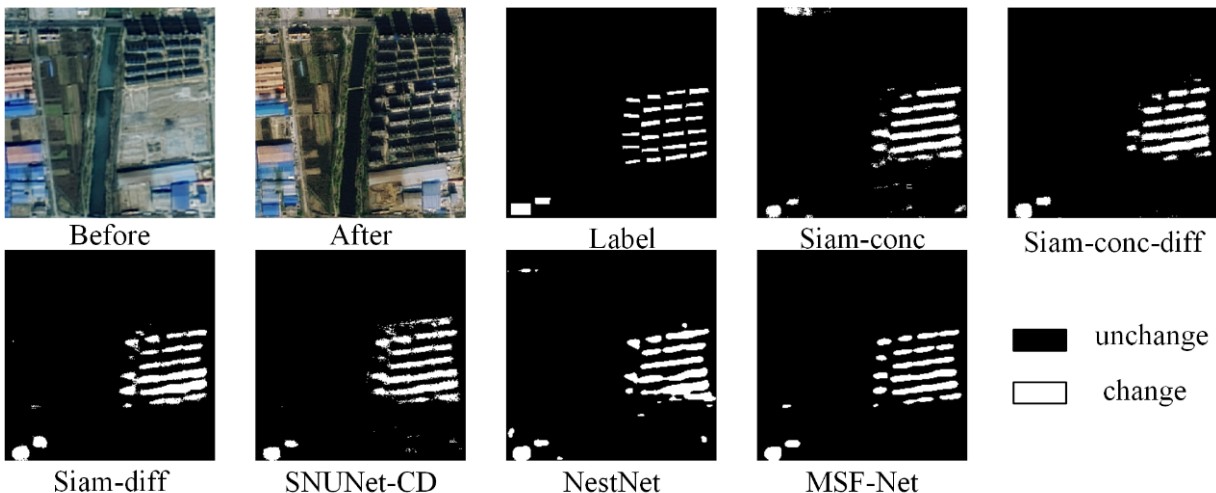

**Figure 8.** Building change detection results of single-source high-resolution images.

Figure 8 shows that the Siam-conc and SNUNet-CD algorithms have more false detection areas, and the detected change buildings are unclear with more broken edges. The changed buildings detected by Siam-conc-diff and NestNet are not complete enough, and the information extraction ability of changed buildings is not enough. The detection results of Siam-diff are slightly better than those of the remaining four advanced algorithms, but the detected building boundaries are still incomplete and blurred. It can be seen that

MSF-Net improves compared with other models, the false detection phenomenon is lower, the detected changed buildings are more complete, and the boundaries are clearer.

### 4.2. Single-Source Multispectral Remote Sensing Images Building Change Detection

Using the annotated MS-HS BCD dataset, building change detection of single-source multispectral images are investigated and experiments are conducted using six network models based on the four multispectral band combinations designed (Tables 4–7). The training loss curves for the six network structures are shown in Figure 9; NestNet and MSF-Net remained flat after epoch60 and the other models remained flat after epoch50. The detection results are shown in Tables 13–18.

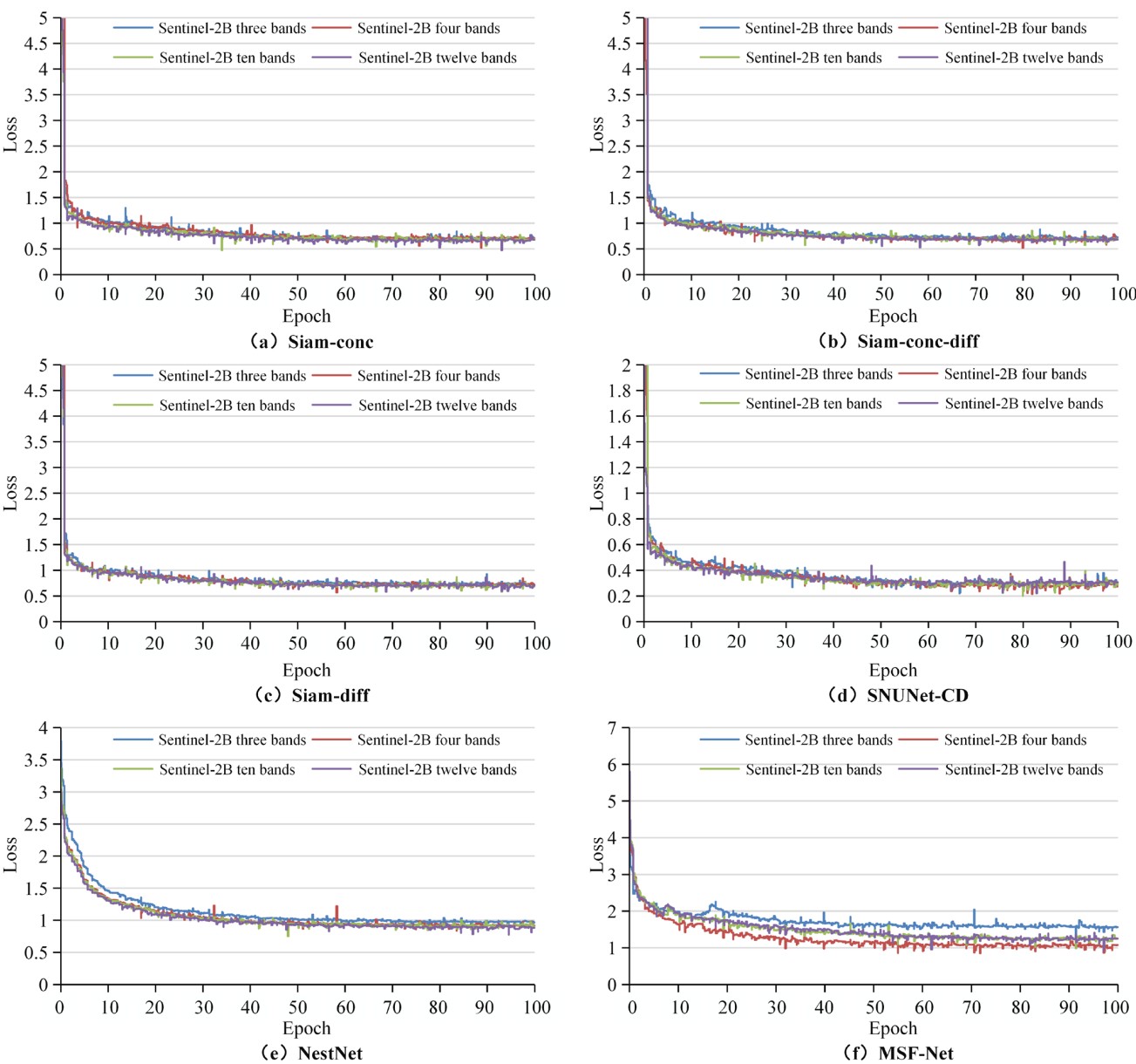

**Figure 9.** Training loss of single-source multispectral remote sensing images.

Table 13 shows the experimental results of the Siam-conc network under four multispectral image data combinations; it works best when the input data are in three bands, with the F1-score and IOU reaching 47.79% and 33.98%, respectively. The network becomes less effective after adding spectral features in multiple bands, indicating that this network

cannot utilize more spectral features well, and the addition of low-resolution spectral features reduces the network effectiveness.

**Table 13.** Siam-conc building change detection results of multispectral images.

| Band | Precision | Recall | F1-Score | IOU |
|---|---|---|---|---|
| Sentinel-2B three bands (Table 4) | 48.76% | 55.96% | 47.79% | 33.98% |
| Sentinel-2B four bands (Table 5) | 46.4% | 53.47% | 43.44% | 30.49% |
| Sentinel-2B ten bands (Table 6) | 45.98% | 58.84% | 46.06% | 31.95% |
| Sentinel-2B twelve bands (Table 7) | 39.84% | 62.81% | 44.09% | 30.5% |

**Table 14.** Siam-conc-diff building change detection results of multispectral images.

| Band | Precision | Recall | F1-Score | IOU |
|---|---|---|---|---|
| Sentinel-2B three bands (Table 4) | 49.82% | 52.36% | 47.89% | 34.46% |
| Sentinel-2B four bands (Table 5) | 43.33% | 59.42% | 45.48% | 32.25% |
| Sentinel-2B ten bands (Table 6) | 45.43% | 55.34% | 45.79% | 32.37% |
| Sentinel-2B twelve bands (Table 7) | 44.17% | 63.17% | 45.24% | 31.64% |

Table 14 shows the experimental results of the Siam-conc-diff network under four multispectral image data combinations; it can be concluded that the Siam-conc-diff model similarly achieves the best results for three-band input data, with the F1-score and IOU reaching 47.89% and 34.46%, respectively, and the low-resolution spectral features reduce the network detection when the spectral bands of the input data increase.

**Table 15.** Siam-diff building change detection results of multispectral images.

| Band | Precision | Recall | F1-Score | IOU |
|---|---|---|---|---|
| Sentinel-2B three bands (Table 4) | 44.76% | 58.24% | 46.64% | 33.81% |
| Sentinel-2B four bands (Table 5) | 44.55% | 57.73% | 44.16% | 31.22% |
| Sentinel-2B ten bands (Table 6) | 48.09% | 59.65% | 49.51% | 35.15% |
| Sentinel-2B twelve bands (Table 7) | 43.17% | 56.22% | 42.56% | 29.64% |

Table 15 shows the experimental results of the Siam-diff network under four multispectral image data combinations; the Siam-diff network performs best when the number of input bands is ten and the precision, recall, F1-score, and IOU are all the highest, reaching 48.09%, 59.65%, 49.51%, and 35.15%, respectively. This indicates that after increasing the spectral bands, Siam-diff learns more spectral features and improves the network detection effectiveness, but the network effectiveness decreases after continuing to increase the 60 m/pixel resolution band, indicating that the too low-resolution affects the network detection effect.

**Table 16.** SNUNet-CD building change detection results of multispectral images.

| Band | Precision | Recall | F1-Score | IOU |
|---|---|---|---|---|
| Sentinel-2B three bands (Table 4) | 49.53% | 55.27% | 48.41% | 34.95% |
| Sentinel-2B four bands (Table 5) | 47.44% | 62.95% | 48.77% | 34.88% |
| Sentinel-2B ten bands (Table 6) | 51.14% | 52.24% | 44.8% | 30.95% |
| Sentinel-2B twelve bands (Table 7) | 50.82% | 58.91% | 50.08% | 36.19% |

Table 16 shows the experimental results of the SNUNet-CD network under four multispectral image data combinations; SNUNet-CD achieves optimal results when the input data are twelve bands, with the F1-score and IOU reaching 50.08% and 36.19%, respectively. This indicates that the attention mechanism designed at the output side can acquire building change characteristics at different scales and improve the detection effectiveness after inputting data from multiple resolution bands.

**Table 17.** NestNet building change detection results of multispectral images.

| Band | Precision | Recall | F1-Score | IOU |
|---|---|---|---|---|
| Sentinel-2B three bands (Table 4) | 49.37% | 58.92% | 49.73% | 35.84% |
| Sentinel-2B four bands (Table 5) | 45.02% | 56.99% | 45.63% | 32.75% |
| Sentinel-2B ten bands (Table 6) | 43.83% | 56.76% | 42.8% | 30.13% |
| Sentinel-2B twelve bands (Table 7) | 47.29% | 60.22% | 44.79% | 31.74% |

Table 17 shows the experimental results of the NestNet network under four multispectral image data combinations; NestNet achieves the best results with only three bands, with the F1-score and IOU reaching 49.73% and 35.84%, respectively. Adding more data bands, the network became less effective, indicating that this network is not capable of handling more data bands.

**Table 18.** MSF-Net building change detection results of multispectral images.

| Band | Precision | Recall | F1-Score | IOU |
|---|---|---|---|---|
| Sentinel-2B three bands (Table 4) | 51.78% | 55.6% | 48.75% | 35.43% |
| Sentinel-2B four bands (Table 5) | 53.76% | 58.4% | 51.65% | 38.19% |
| Sentinel-2B ten bands (Table 6) | 51.85% | 59.32% | 50.13% | 36.37% |
| Sentinel-2B twelve bands (Table 7) | 40.16% | 61.81% | 43.69% | 30.62% |

Table 18 shows the experimental results of the MSF-Net network under four multispectral image data combinations; MSF-Net works best at four bands, with the F1-score and IOU reaching 51.65% and 38.19%, respectively. The detection becomes less effective after continuing to include band data with resolutions of 20 m/pixel and 60 m/pixel, indicating that the MSF-Net algorithm can utilize more spectral features with higher resolutions and is less capable of processing spectral information with lower resolutions.

The performance of the six deep neural networks on the single-source multispectral remote sensing image building change detection dataset shows that Siam-conc, Siam-conc-diff, and NestNet achieve optimal results when the number of input bands is three, while Siam-diff, SNUNet-CD, and MSF-Net achieve optimal results when the number of input bands is 10, 12, and 4, respectively. This reflects the different learning abilities of different networks for different data bands. Among the six algorithms, MSF-Net has the best detection effect when using four-band multispectral image data as the input and improves the F1-score by 1.57% and the IOU by 2% compared with SNUNet-CD, which has the best performance among the remaining algorithms using 12 bands as the data input.

*4.3. Multisource Spectral and Texture Feature Building Change Detection*

Using six network models based on the MS-HS BCD dataset, experiments on building change detection in multisource spectral and texture features are conducted based on three different band combinations of high-resolution and multispectral images. We compared the experimental results of the three combined multisource data, the single-source high-resolution images experimental results, and the results for the single-source multispectral images. It is important to note that, among the results for single-source multispectral data, the Siam-conc, Siam-conc-diff, Siam-diff, SNUNet-CD, NestNet, and MSF-Net networks achieve the best results at three, three, ten, twelve, three, and four bands, respectively, so only these data combinations are used for the comparison of multisource data. The losses of the six network models on the three multisource data combinations are shown in Figure 10; the losses of the six network models remained flat after epoch50, the multisource data combination method had a greater impact on the MSF-Net, and the loss is significantly reduced at the four bands combination. The quantitative comparison results of each network model are shown in Tables 19–24, and the results of building change detection at different data combinations are shown in Figures 11–16.

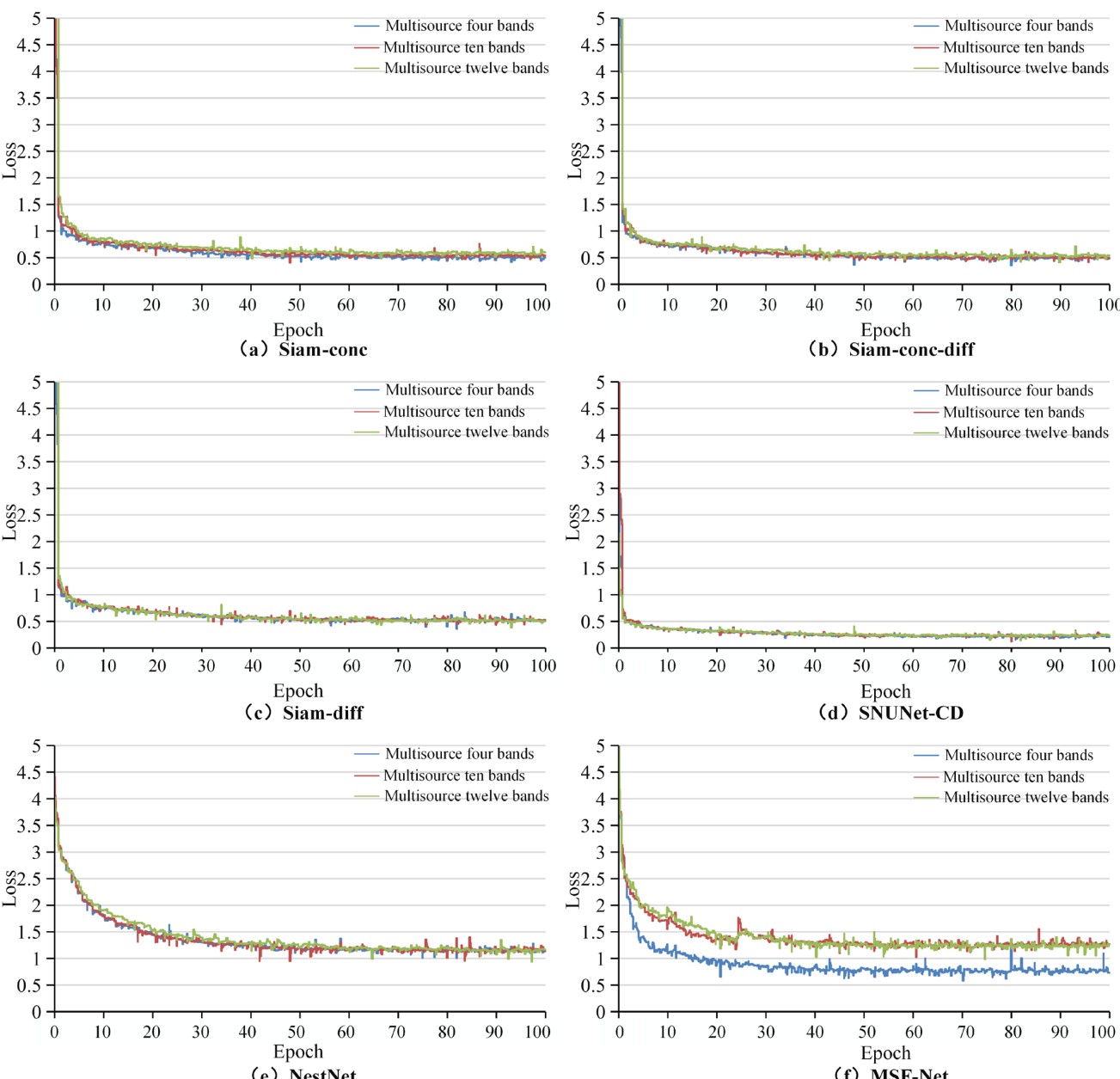

**Figure 10.** Training loss of multisource remote sensing images.

**Table 19.** Siam-conc building change detection results of multisource spectral and texture features.

| Data Source | Precision | Recall | F1-Score | IOU |
|---|---|---|---|---|
| High-resolution (Table 3) | 48.43% | 53.29% | 45.3% | 31.49% |
| Multispectral three bands (Table 4) | 48.76% | 55.96% | 47.79% | 33.98% |
| Multisource four bands (Table 8) | 52.53% | 55.34% | 48.97% | 35.54% |
| Multisource ten bands (Table 9) | 46.99% | 57.46% | 47.23% | 33.68% |
| Multisource twelve bands (Table 10) | 49.66% | 55.67% | 48.65% | 35.08% |

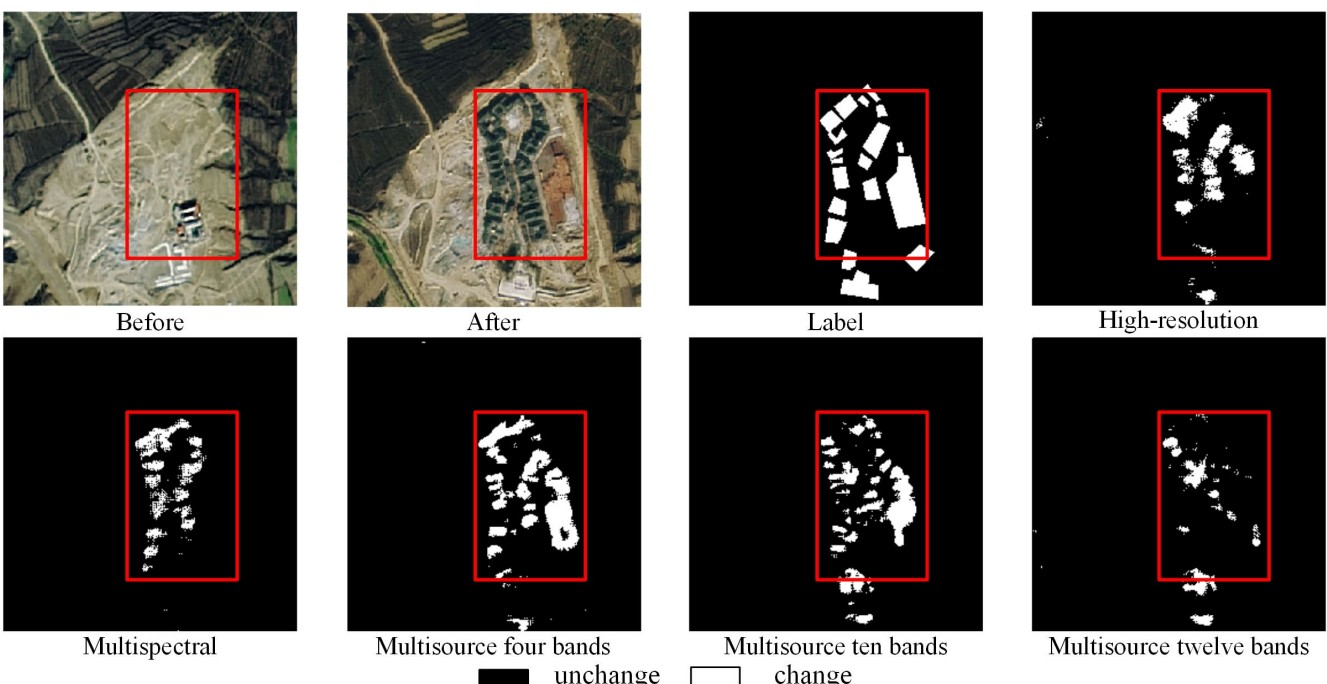

**Figure 11.** The multisource spectral and texture feature detection results of Siam-conc.

Table 19 shows the experimental results of the Siam-conc network under multisource spectral and texture data combinations. The Siam-conc algorithm achieves the best detection results when the four-band multisource data are combined; the precision reaches 52.53%, the F1-score reaches 48.97%, and the IOU reaches 35.54%, which are 4.1%, 3.67%, and 4.05% higher than the high-resolution images and 3.77%, 1.18%, and 1.56% higher than the multispectral images, respectively. Recall is optimal for the ten-band combination of multisource data, which indicates that the detection effectiveness of the Siam-conc algorithm can be improved by adding multisource spectral information with higher resolution. Continuing to add multisource spectral information, the detection effect decreases when the band combination is 10 and 12, indicating that adding too much low-resolution multisource data for this network model has a negative effect and reduces the model detection effectiveness. In Figure 11, when the four-band multisource data were combined, Siam-conc detected changed buildings more completely and clearly.

Table 20 shows the experimental results of the Siam-conc-diff network under multisource spectral and texture data combinations. The Siam-conc-diff algorithm achieves the best detection at the ten-band combination, with recall reaching 62.22%, F1-score reaching 55.02%, and IOU reaching 40.09%, improving by 8.97%, 7.96%, and 6.1%, respectively, compared to the high-resolution images, and 9.86%, 7.13%, and 5.63%, respectively, compared to multispectral images. Precision reached the highest at 55.37% for the four-band combination. This indicates that the algorithm can better utilize more spectral information to improve detection precision after merging the dual-temporal image features and their differential features. The detection results of twelve-band data decreased significantly compared with those of ten-band data, indicating that adding low-resolution band data has a greater negative impact on the Siam-conc-diff network. Figure 12 shows that when ten multisource spectral bands are combined, Siam-conc-diff can detect changed buildings more precisely.

**Table 20.** Siam-conc-diff building change detection results of multisource spectral and texture features.

| Data Source | Precision | Recall | F1-Score | IOU |
|---|---|---|---|---|
| High-resolution (Table 3) | 54.12% | 53.25% | 47.06% | 33.99% |
| Multispectral three bands (Table 4) | 49.82% | 52.36% | 47.89% | 34.46% |
| Multisource four bands (Table 8) | 55.37% | 58.32% | 51.85% | 37.54% |
| Multisource ten bands (Table 9) | 54.35% | 62.22% | 55.02% | 40.09% |
| Multisource twelve bands (Table 10) | 50.88% | 58.78% | 51.24% | 36.99% |

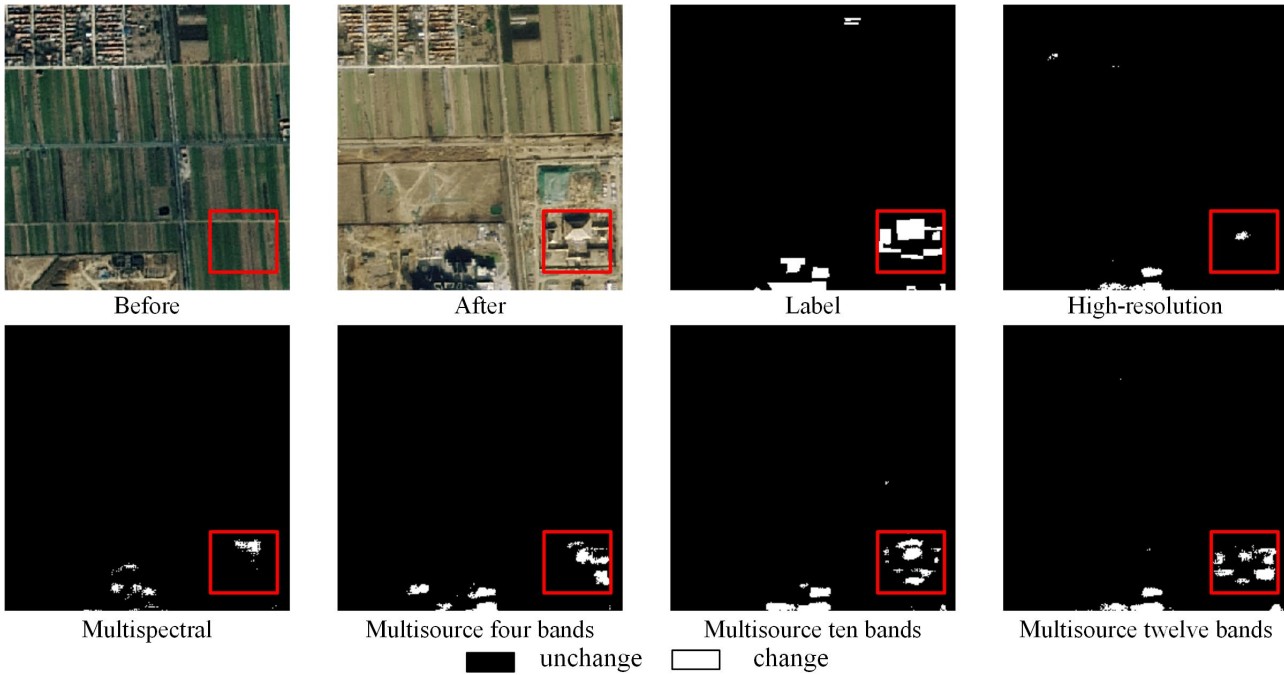

**Figure 12.** The multisource spectral and texture feature detection results of Siam-conc-diff.

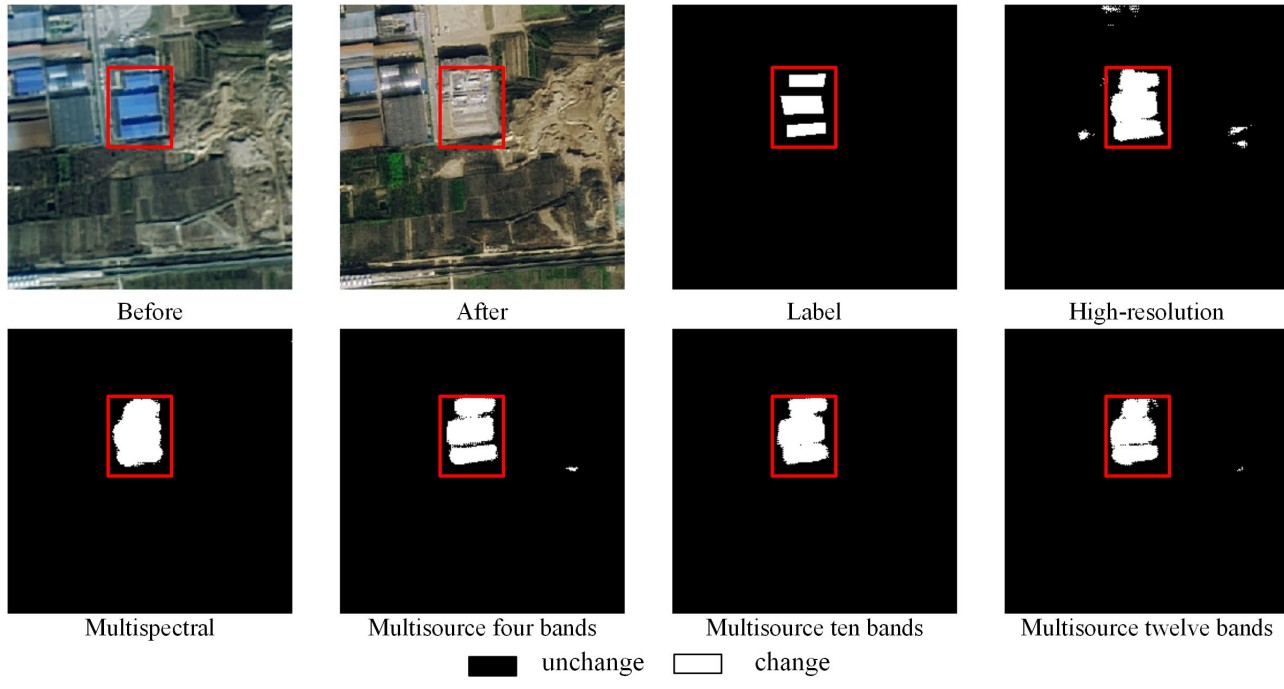

**Figure 13.** The multisource spectral and texture feature detection results of Siam-diff.

Table 21 shows the experimental results of the Siam-diff network under multisource spectral and texture data combinations. The Siam-diff algorithm has the best detection effect in the four-band combination; although the recall decreases, the precision reaches 62.76%, the F1-score reaches 52.64%, and the IOU reaches 38.28%. The F1-score improves by 4.62% and 3.13% compared with the high-resolution image and multispectral image, respectively, and the IOU improves by 3.21% and 3.13%, respectively. The F1-score and IOU of the model continue to decrease as we continue to add multisource spectral information, indicating that the Siam-diff network can use higher resolution multisource image data to improve the detection precision but cannot handle more low-resolution spectral data. Figure 13 shows that when four multisource spectral bands are combined, Siam-diff can detect clearer building boundaries.

**Table 21.** Siam-diff building change detection results of multisource spectral and texture features.

| Data Source | Precision | Recall | F1-Score | IOU |
|---|---|---|---|---|
| High-resolution (Table 3) | 50.08% | 54.77% | 48.02% | 35.07% |
| Multispectral ten bands (Table 6) | 48.09% | 59.65% | 49.51% | 35.15% |
| Multisource four bands (Table 8) | 62.76% | 52.19% | 52.64% | 38.28% |
| Multisource ten bands (Table 9) | 52.98% | 54.51% | 50.22% | 36.93% |
| Multisource twelve bands (Table 10) | 54.88% | 51.19% | 49.48% | 35.49% |

**Table 22.** SNUNet-CD building change detection results of multisource spectral and texture features.

| Data Source | Precision | Recall | F1-Score | IOU |
|---|---|---|---|---|
| High-resolution (Table 3) | 43.42% | 55.22% | 44.68% | 31.83% |
| Multispectral twelve bands (Table 7) | 50.82% | 58.91% | 50.08% | 36.19% |
| Multisource four bands (Table 8) | 52.26% | 56.64% | 50.43% | 36.41% |
| Multisource ten bands (Table 9) | 48.59% | 54.41% | 46.4% | 33.09% |
| Multisource twelve bands (Table 10) | 53.72% | 55.48% | 47.68% | 33.75% |

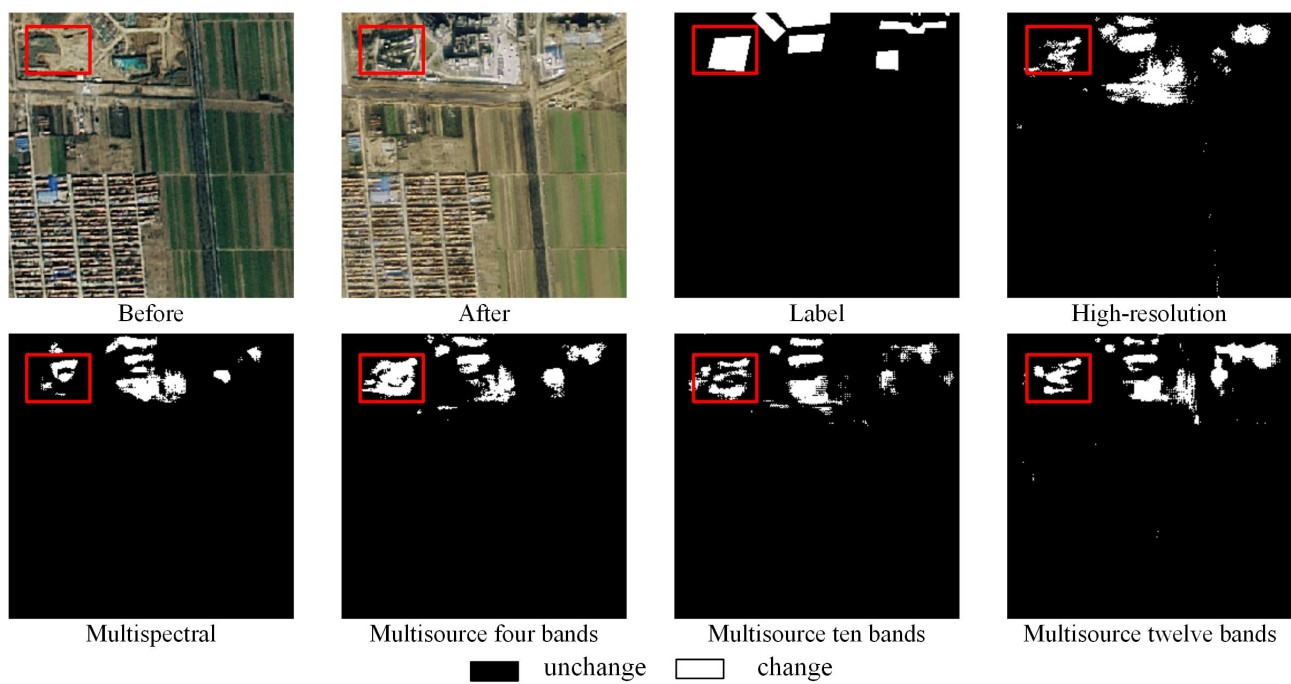

**Figure 14.** The multisource spectral and texture features detection results of SNUNet-CD.

Table 22 shows the experimental results of the SNUNet-CD network under multisource spectral and texture data combinations. From the two important metrics, F1-score and IOU, the SNUNet-CD algorithm works best in the four-band combination of multisource

data, with the F1-score and IOU reaching 50.43% and 36.41%, respectively. However, the network is weak in extracting information from multisource image data, and although the two metrics, the F1-score and IOU, improve by 5.75% and 4.58%, respectively, compared with high-resolution images, they only improve by 0.35% and 0.22% compared with single-source multispectral images. These results indicate that the SNUNet-CD network achieves better results in extracting multispectral image features. Adding more multisource data information does not significantly, improve its effect and the effect of the network becomes worse when adding more low-resolution multisource spectral features. In Figure 14, when four multisource spectral bands are combined, SNUNet-CD can detect changed buildings more completely.

**Table 23.** NestNet building change detection results of multisource spectral and texture features.

| Data Source | Precision | Recall | F1-Score | IOU |
|---|---|---|---|---|
| High-resolution (Table 3) | 41.48% | 60.46% | 43.84% | 30.96% |
| Multispectral three bands (Table 4) | 49.37% | 58.92% | 49.73% | 35.84% |
| Multisource four bands (Table 8) | 53.8% | 57.85% | 50.72% | 36.45% |
| Multisource ten bands (Table 9) | 47.34% | 61.23% | 50.03% | 35.99% |
| Multisource twelve bands (Table 10) | 51.62% | 58.72% | 50.92% | 36.78% |

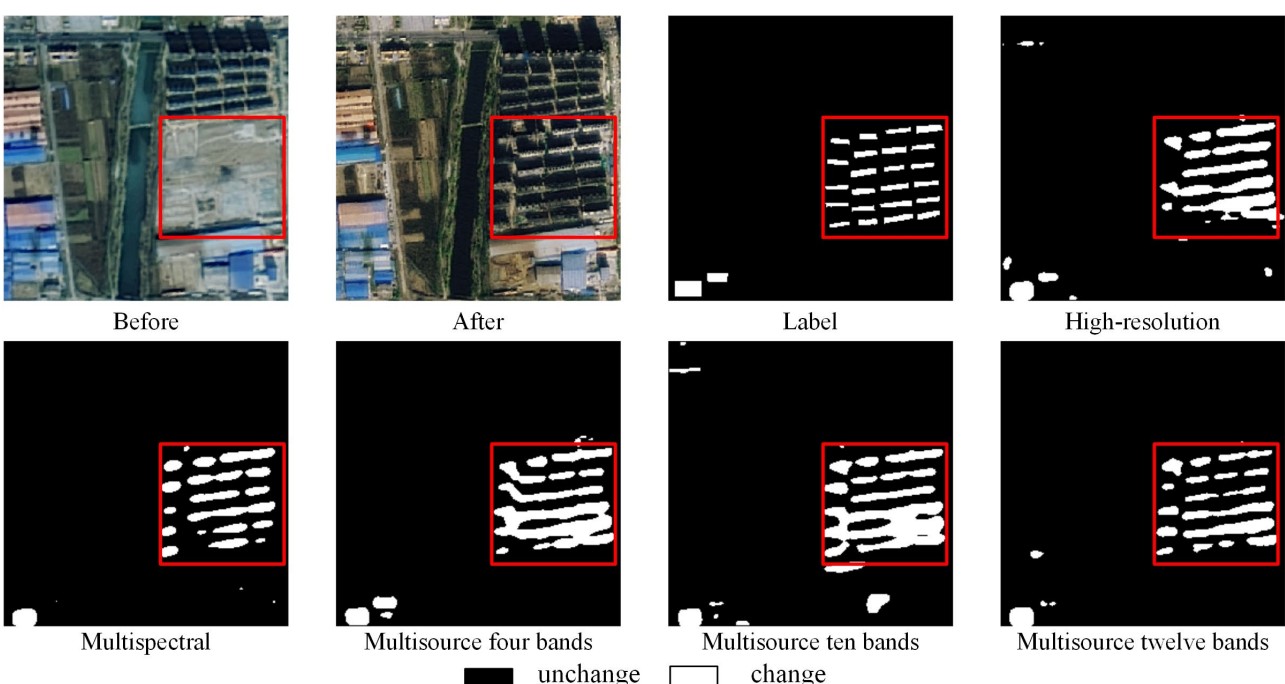

**Figure 15.** The multisource spectral and texture feature detection results of NestNet.

Table 23 shows the experimental results of the NestNet network under multisource spectral and texture data combinations. The F1-score and IOU, two important metrics of NestNet, achieved the best results when the twelve-band multisource image data were combined, reaching 50.92% and 36.78%, respectively. Precision was the highest at 53.8% for the four-band combination. Recall was the highest at 61.23% for the ten-band combination. The four evaluation indicators showed a significant increase compared with single-source images. Thanks to the improved UNet++ dense skip connection module, the NestNet algorithm can use more multisource spectral data to effectively improve the building change detection effect compared with single-source remote sensing image data. In Figure 15, it can be seen that NestNet works best when twelve multisource spectral bands are combined and the boundaries of the changing buildings are clearer.

**Table 24.** MSF-Net building change detection results of multisource spectral and texture features.

| Data Source | Precision | Recall | F1-Score | IOU |
|---|---|---|---|---|
| High-resolution (Table 3) | 61.1% | 65.02% | 58.55% | 43.31% |
| Multispectral four bands (Table 5) | 53.76% | 58.4% | 51.65% | 38.19% |
| Multisource four bands (Table 8) | 61.63% | 64.41% | 59.22% | 44.4% |
| Multisource ten bands (Table 9) | 54.07% | 52.53% | 48.98% | 36.41% |
| Multisource twelve bands (Table 10) | 52.25% | 58.9% | 50.67% | 36.54% |

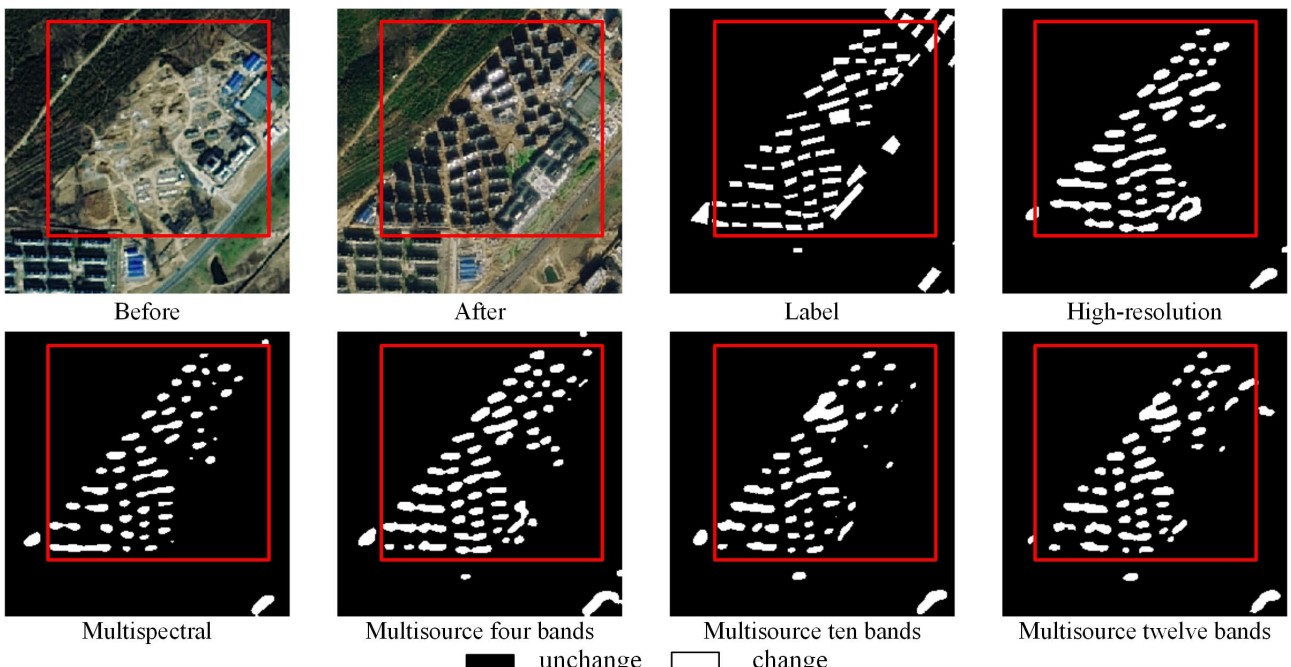

**Figure 16.** The multisource spectral and texture feature detection results of MSF-Net.

Table 24 shows the experimental results of the MSF-Net network under multisource spectral and texture data combinations. MSF-Net works best with the combination of four-band multisource data. Although recall decreases by 0.61% compared with single-source high-resolution images, the precision, F1-score, and IOU improve by 61.63%, 59.22%, and 44.4%, respectively. This indicates that adding multisource spectral information with higher resolution data can make MSF-Net learn more building features and improve the detection effectiveness, and the model detection becomes less effective when continuing to add multisource spectral information, indicating that the addition of too much low-resolution data affects the algorithm performance. Figure 16 shows the building change detection results under different band combinations. When the four-band multisource data were combined, the changing buildings were detected more completely and with clearer boundaries. This indicates that MSF-Net can use more spectral features and texture information to improve the effect of building change detection.

In order to more significantly represent the variation in each model evaluation metric with data source, we organized the results of the Tables 19–24 into a bar chart, as shown in Figure 17. It is important to note that the multispectral combinations in the Siam-conc, Siam-conc-diff, Siam-diff, SNUNet-CD, NestNet, and MSF-Net are three-, three-, ten-, twelve-, three-, and four-bands combination, respectively. The data combinations of high-resolution, multisource four bands, multisource ten bands, multisource twelve bands are shown in Tables 3 and 8–10. It can be concluded that building the change detection based on multisource spectral and texture feature data can effectively improve the detection effect of the algorithm model. Except for the decrease in recall of the Siam-diff, SNUNet-CD. and MSF-Net algorithms, all of the other indices increase to different degrees.

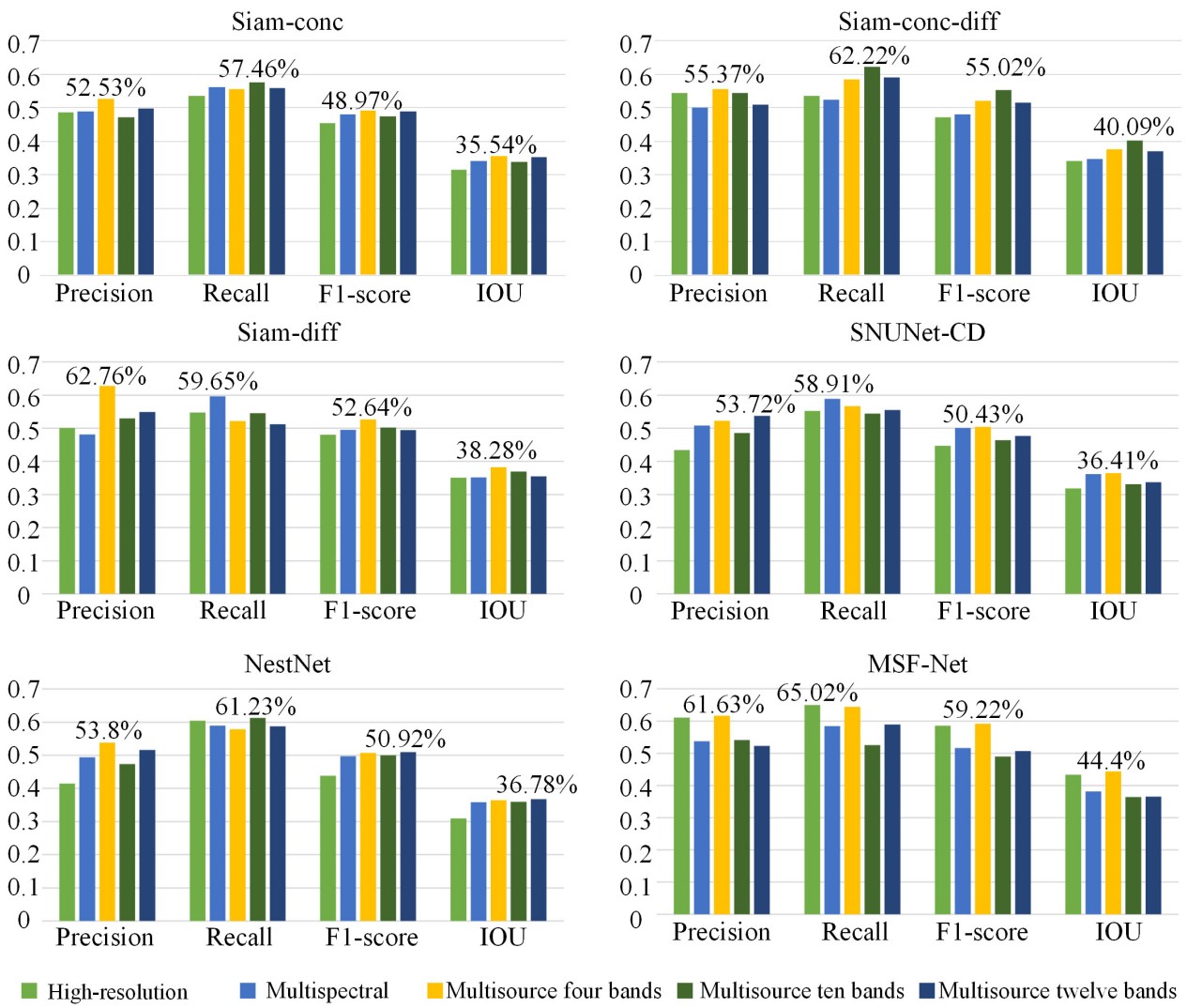

**Figure 17.** The evaluation index of each model changes with the data source.

## 5. Conclusions

In this paper, a multisource remote sensing image building change detection dataset (MS-HS BCD dataset) is produced based on two kinds of images, GF-1 and Sentinel-2B, and three kinds of data combinations are classified: single-source high-resolution data, single-source multispectral data with four-band combination methods, and multisource data with three-band combination methods. Based on multiple data combination methods, six state-of-the-art change detection neural network models are selected for building change detection experiments with multisource spectral and texture feature data. The experimental results show the following: (1) When inputting multisource spectral and texture feature data into the neural network, all six network models achieve different degrees of improvement, among which SNUNet-CD has the smallest improvement and the F1-score and IOU have less than 1% improvement compared with single-source multispectral images. The improvement of Siam-conc-diff is the largest and the improvement of the F1-score and IOU is 7.13% and 5.63%, respectively, compared with single-source multispectral images, indicating that different network models have different learning abilities for multisource data. (2) Siam-conc, Siam-diff, SNUNet-CD, and MSF-Net achieve the best results when the four-band data are combined, and Siam-conc-diff and NestNet achieve the best results when the ten-band and twelve-band data are combined. The detection effect of some models decreases after adding lower resolution band data, indicating that the excessive addition of multisource spectral feature data with lower resolutions will reduce the effectiveness

of the model. Combining the above experimental results, it can be concluded that the performance of the model is improved compared with the results based on single-source data when multiple sources are fed into the model simultaneously, indicating that the lack of single-source high-resolution spectral features or the coarse texture features of single-source multispectral images limit the learning of changing building features by the network model. By combining them, the network model can combine the more spectral and fine texture features of these data sources to enhance the learning ability of building features so that the model can detect more complete changing buildings and improve the detection effect.

In summary, our paper firstly proposed an open-source multisource building change detection dataset, which provides a database for multisource building change detection research and makes up for the lack of such datasets. Next, the effect of multisource data on the building detection effect was performed based on this dataset. The experimental results showed that the detection effect can be significantly improved when multisource spectral and texture features were simultaneously inputted to the model, and the research in this paper also provided a reference for the continued exploration in this field.

Although we obtained some research results in building change detection in multi-source remote sensing data, there are still some limitations in the current research. The multisource building change detection dataset (MS-HS BCD dataset) produced in this paper was a small volume dataset with only 600 image sets, which may affect the detection effectiveness of the model to some extent. The preprocessing method of dataset can be further improved. For example, our future research direction includes exploring the influence of resampling with higher resolution (2 m) on the experimental results and exploring the geographical registration methods of two kinds of images with higher accuracy.

Our future work will also introduce radar, SHP vector data, etc., to explore the impact of more types of data sources on building change detection.

**Author Contributions:** Conceptualization, J.F. and J.C.; Data curation, M.Z. and J.Z.; Formal analysis, M.Z., J.C., J.Z. and Z.S.; Funding acquisition, J.F.; Methodology, J.F.; Software, M.Z., J.Z. and Z.S.; Supervision, J.F. and M.J.; Visualization, M.Z.; Writing–original draft, J.F. and J.C.; Writing–review and editing, J.F. and M.J. All authors have read and agreed to the published version of the manuscript.

**Funding:** This work was supported by the National Natural Science Foundation of China (Grant No. 42171413); a grant from the State Key Laboratory of Resources and Environmental Information System; the Shandong Provincial Natural Science Foundation (Grant No. ZR2020MD015 and ZR2020MD018); the National Key Research and Development Program of China (Grant No. 2017YFB0503500); and the Young Teacher Development Support Program of Shandong University of Technology (Grant No. 4072-115016).

**Data Availability Statement:** Our proposed dataset will be released through GitHub (https://github.com/arcgislearner/MS-HS-BCD-dataset (accessed on 6 April 2023)).

**Acknowledgments:** We would like to thank the Shandong University of Science and Technology, ESA Copernicus Data Center for access to the data. We appreciate the editors and reviewers for their constructive comments and suggestions.

**Conflicts of Interest:** The authors declare no conflict of interest.

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
