# Peer review of "Building Change Detection with Deep Learning by Fusing Spectral and Texture Features of Multisource Remote Sensing Images: A GF-1 and Sentinel 2B Data Case"

_remotesensing, doi:10.3390/rs15092351_

Round 1

Reviewer 1 Report

In this study, authors produced a multisource building change detection dataset named (MS-HS BCD) based on GF-1 and Sentinel-2B images. Further authors used this dataset and compared six state-of-the-art deep learning models for building change detection.

Following are the suggestions to improve the final manuscript:

·         Page number 1, lines 34-36 (sentence starting with “Before….”), rewrite the sentence.

·         Page number 2, line number 47, full stop missing.

·         Line number 88, spelling mistake.

·         Page number 4, line number 148, it is mentioned that “Sentinel-2B images were resampled to 2.5 m/pixel….”. How this resolution enhancement was performed?

·         Page number 6, section 3.2, it is not clear from the text which bands were used in each set. It is suggested that for each combination make a table that should include the band name and wavelength of the band used.

·         It is suggested to make a table of the experimental environment.

·         Tables 6-18 are not cited in the text. It is suggested to cite the table in the text and mention what the table is representing.

·         In section 4.2, which dataset was used is not clear.

·         Table 13 onwards, in the first two rows, which data/band combination was used, is not clear.

·         From the manuscript, it seems that authors have not trained different models used in the study using MS-HS BCD data. If trained then the training/validation loss curve should be added, if not then it is suggested to train the models on MS-HS BCD data.

Author Response

Dear reviewer, 
Thank you very much for your valuable revision comments, which point out the errors and deficiencies in the research and writing, and I have benefited greatly. In the following, I will respond item by item. I have uploaded the response file in which your comments were responsed item by item. Thank you again!

Reviewer 2 Report

In this paper using GF-1 high-resolution 20 remote sensing images and Sentinel-2B multispectral remote sensing images. To investigate the influence of spectral features on the effect of building change detection based on deep learning, a multisource building change detection dataset (MS-HS BCD dataset) is discussed in the paper.

Q1. In the proposed model which has almost few basic theoretical, it is suggested the applied background should be discussed in-depth. What’s the reason, why use it? and so on.

Q2. The authors should explain why they adopted the proposed models for their applications. Are they all the suitable adoption for the evaluation? What’re the practice contribution results from the article?

Q3. The authors should give graphical models to optimize the applied models. Moreover, the authors should give the novel mathematical models to prove the performance of those models.

Q4. The authors should explain why they didn't adopt other models. The authors should compare the results with many existed studies.

Q5. The motivation and solving problems should give in-depth discussed, thought huge amount of dataset are applied in the investigation.

Q6. The authors should propose an original method to enhance the level of contributions, if possible.

Q7. The authors should highlight the contributions of this study in the first section.

Q8. The guarantee of the obtained data should be provided in advances, especially the one shown in Figure 10.

Author Response

(The authors gave the same response as above.)

Reviewer 3 Report

You used the GF-1 image in the experiment, but you only used three RGB bands. It’s a little odd that, since GF-1 has four bands including NIR, why not exclude it in your analysis?

You have resampled the 2 m GF-1 to 2.5m and also resampled the 10 m Sentinel-2 bands to 2.5m. What method did you use when resampling the GF-1 image (this could change the DN or reflectance values for GF-1)? It’s odd that why not resample the 10 m S-2 image to 5 m?

How do you co-registrate the GF-1 and Sentinel-2 bands in your analysis?

You have used 12 S-2 bands, and I’m wondering if it is necessary, because the 60 m bands may be not sensitive to the buildings. Explain the reason for your processing.

A comparison method using only Sentinel-2 image is needed.

It is necessary to list the acquisition date of the GF-1 and Sentinel-2 imagery. If these data are acquired on different dates, the buildings may change and the combination is thus unsuitable.

Did you augment your data for training? You highlighted the training data may be not sufficient.

The conclusion section should be enhanced with more theoretical analysis. 

Author Response

(The authors gave the same response as above.)

Round 2

Reviewer 1 Report

The authors have addressed all comments/suggestions

Author Response

The responses was included in the uploaded file. Thank you!

Reviewer 3 Report

The authors have replied to my responses. Some of them should be revised. For instance, I agree resampling 10 m to 2.5 m using the nearest neighbor sampling does not change the DN value, but the coarsening of GF-1 will result in a loss of information. So why do you resample GF-1? You can resample Sentinel-2 to 2 m (not 2.5 m) and this process will not degrade the GF-1 data. The co-registration error is also not provided. I think these issues should be highlighted to make this paper clear. 

Author Response

(The authors gave the same response as above.)
